# High-resolution biomechanical mapping of SMILE and SMILE with CXL using Brillouin microscopy: Insights into localized corneal stiffness preservation

Jiwon Jeong[1,2]*, Mincheol Bae[1], Dong Wook Kim[3], Hui June Kim[1], Younghee Kim[2]

**1** Fatima Eye Clinic, Changwon, Republic of Korea, **2** Fatima Eye Research, Changwon, Republic of Korea, **3** Department of Information and Statistics, Department of Bio & Medical Big Data, Research Institute of Natural Science, Gyeongsang National University, Jinju, Republic of Korea

* jjwdoc@naver.com

## Abstract

### Purpose

To evaluate the biomechanical effects of small incision lenticule extraction (SMILE) alone and with corneal collagen cross-linking (SMILE$_{CXL}$) on corneal stiffness, using the Brillouin Optical Scanning System (BOSS) to assess localized changes in Brillouin modulus (BM).

### Materials & Methods

This retrospective study analyzed an initial cohort of 358 eyes (321 SMILE, 37 SMILE$_{CXL}$) from 180 patients, with propensity score matching (PSM) yielding matched cohorts of 17 eyes each, including unilateral and bilateral cases. PSM balanced baseline characteristics using 10 variables: age, gender, lenticular thickness, residual stromal thickness, spherical power, cylinder, spherical equivalent, flat keratometry, steep keratometry, and mean keratometry. To address the violation of independence in the between-group comparison of pre- and postoperative ΔBM due to the inclusion of bilateral eyes from some participants, generalized estimating equations (GEE) with an exchangeable correlation structure were used to account for within-person correlation in clustered data from bilateral eyes. BOSS (Intelon Optics, Woburn, MA, USA) measured BM via Brillouin frequency shifts (gigapascals, GPa) at 10 predefined corneal points (central, superior, inferior, nasal, temporal; coordinates −2.4 mm to +2.4 mm from apex) preoperatively and at 1 and 3 months postoperatively. Post-PSM, BM values and changes (ΔBM) were compared, including point-specific, mean, minimum, and maximum analyses. ΔBM patterns (≥ 0 for preservation/increase, $p < 0.05$ for inter-group significance) were categorized for regional insights.

**Data availability statement:** Data cannot be shared publicly due to legal restrictions under South Korean law, including the Personal Information Protection Act, and hospital policies at Fatima Eye Clinic, which prohibit the external sharing of clinical patient data, even when anonymized, without additional ethical review. Data are available upon reasonable request from the Fatima Eye Clinic Data Access Committee (fatimaeye@naver.com), subject to institutional approval for researchers who meet the criteria for access to confidential data.

**Funding:** The author(s) received no specific funding for this work.

**Competing interests:** The authors have declared that no competing interests exist.

## Results

Pre-PSM differences in age, lenticular thickness, residual stromal thickness, and refractive parameters (e.g., spherical equivalent) necessitated matching; post-PSM, all were balanced ($p > 0.05$). Preoperative BM values were comparable. Post-SMILE, stiffness decreased (mean $\Delta$BM −0.021 GPa at 1 month, −0.009 GPa at 3 months), with partial recovery but 80% points below baseline; BM4/BM9 spontaneously recovered ($\Delta$BM ≥ 0 at 3 months). SMILE$_{CXL}$ preserved/enhanced stiffness across all points (mean $\Delta$BM +0.015 GPa at 1 month, +0.011 GPa at 3 months), with significant differences at minimum $\Delta$BM ($p < 0.05$ at 1 month) and points BM1, BM4, BM5 ($p < 0.05$ at 1 month), BM1 ($p < 0.05$ at 3 months). Four patterns emerged: (1) Resistant but potentially vulnerable (BM7/BM10: near-zero initial loss but worsening, inferior progressive loss); (2) Spontaneously and fully recovered (BM4/BM9: initial loss but recovery ≥ 0 at 3 months without CXL, superior resilience); (3) Moderately effective for CXL (BM2/BM3/BM6/BM8: loss with non-significant preservation, peripheral benefits); (4) Highly effective for CXL (BM1/BM5: pronounced loss but significant preservation [$p < 0.05$], nasal-temporal central protection).

## Conclusions

BOSS's high-resolution spatiotemporal analysis reveals that CXL counters SMILE-induced stiffness loss, achieving universal preservation or enhancement across corneal zones ($\Delta$BM ≥ 0 at all points and timepoints) and significant protection in vulnerable nasal-temporal central areas ($p < 0.05$). These patterns–inherent superior resilience, inferior susceptibility, zone-specific CXL efficacy–suggest BOSS's potential utility in mapping localized biomechanical changes, offering insights for monitoring postoperative stiffness in refractive procedures.

## Introduction

The cornea's biomechanical integrity is paramount for preserving structural stability, optical clarity, and long-term ocular health, as deviations in stiffness can induce surface irregularities, diminished transparency, and progressive complications like ectasia [1]. Refractive surgeries, such as laser in situ keratomileusis (LASIK), photorefractive keratectomy (PRK), and small incision lenticule extraction (SMILE), are widely adopted for correcting myopia, hyperopia, and astigmatism [2,3]. However, these procedures can weaken corneal biomechanics, potentially leading to visual distortion, structural instability, or progressive ectasia if changes persist, particularly in patients with predisposing traits such as high myopia or thin corneas [4,5]. This underscores the imperative for sophisticated evaluations of corneal mechanics to preempt risks, fueling innovations in strategies that bolster stability while safeguarding refractive efficacy.

Among refractive modalities, SMILE distinguishes itself through its flapless, minimally invasive technique, which conserves anterior corneal layers and yields enhanced biomechanical retention relative to flap-dependent methods, as substantiated by finite element models showing reduced stress and strain distributions [6,7]. Corneal collagen cross-linking (CXL) has emerged as a promising method to enhance corneal stiffness, mitigating these risks through riboflavin-UVA-induced collagen bonding [8,9]. Recent studies, such as those by Kanellopoulos et al. [10] and Hersh et al. [11], demonstrate that CXL combined with LASIK or PRK improves corneal stability, reducing ectasia risk in high-myopia patients. The amalgamation of SMILE and CXL promises synergistic benefits, merging precise refractive correction with augmented structural reinforcement, particularly advantageous for high-risk demographics susceptible to postoperative instability [12,13].

Conventional instruments for gauging corneal biomechanics, including the Ocular Response Analyzer (ORA) and Corvis ST, utilize air-puff deformation to derive metrics like hysteresis and resistance factor; however, these are constrained by their susceptibility to confounding variables and reliance on composite indices that may obscure intricate biomechanical subtleties [14,15]. Conversely, the Brillouin Optical Scanning System (BOSS) facilitates non-invasive, high-resolution quantification of corneal stiffness through Brillouin frequency shifts, yielding the longitudinal elastic modulus in healthy corneas [typically ~2.80–2.88 gigapascals (GPa)] with associations to variables like age and central thickness [16,17]. BOSS research has illuminated focal biomechanical alterations in refractive scenarios, such as post-LASIK central weakening and CXL-mediated stiffening in keratoconus, underscoring its prowess in detecting localized variations [18,19]. Localized biomechanical measurements with BOSS enable detection of regional stiffness variations, which may inform ectasia risk assessment in high-myopia or thin-cornea patients by identifying vulnerable zones for targeted interventions [4,5]. The accelerated epi-on CXL protocol, involving riboflavin application without epithelial removal followed by high-intensity UVA irradiation, minimizes recovery time and complications while effectively increasing collagen cross-links, as supported by studies demonstrating sustained stiffness gains in keratoconus models [7,8]. Nonetheless, extant studies predominantly emphasize aggregate metrics or central zones, often neglecting spatiotemporal dynamics, regional heterogeneities, and longitudinal pattern analyses, which limits comprehensive insights into zonal vulnerabilities and recovery trajectories [20–24]. Moreover, BOSS applications in SMILE or SMILE with CXL are virtually absent, highlighting a critical gap in high-resolution biomechanical profiling for these procedures.

This study evaluates the biomechanical efficacy of SMILE combined with CXL (SMILE$_{CXL}$) compared to SMILE alone, pioneering the deployment of BOSS to scrutinize corneal stiffness alterations post-SMILE and SMILE$_{CXL}$, harnessing its precision to unravel localized fluctuations across predefined points and emphasizing dynamics in refractive procedures. Through propensity-matched cohort comparisons pre- and postoperatively, we seek to delineate nuanced biomechanical responses via BM assessment, furnishing novel perspectives that propel investigations toward enhanced understanding of refractive surgery outcomes and corneal stability.

## Materials and methods

### Study design and ethics

This retrospective, propensity score-matched study analyzed 358 eyes from 180 patients undergoing SMILE or SMILE$_{CXL}$ for myopia correction at Fatima Eye Clinic (Changwon, Republic of Korea) from September 2023 to December 2024. Data were accessed for research purposes on September 13, 2025 (initial extraction of anonymized medical records from the clinic database). The study was approved by the Public Institutional Review Board of the Korea Ministry of Health and Welfare (https://public.irb.or.kr; P01–202508–01–058), adhering to the Declaration of Helsinki. Patient data were anonymized for confidentiality, and informed consent was waived due to the retrospective design. Authors did not have access to information that could identify individual participants during or after data collection; data were fully de-identified prior to analysis.

## Participants

This initial cohort comprised 321 eyes (161 patients) in the SMILE group and 37 eyes (19 patients) in the SMILE$_{CXL}$ group, with propensity score matching (PSM) yielding matched cohorts of 17 eyes each. PSM balanced baseline characteristics using 10 variables: age, gender, lenticular thickness (LT), residual stromal thickness (RST), spherical power (SPH), cylinder (CYL), spherical equivalent (SE), flat keratometry (K1), steep keratometry (K2), and mean keratometry (Km). All analyses were performed on the post-PS matched cohorts. Inclusion criteria were adults aged 18–40 years with stable myopia (spherical equivalent [SE] −3 to −10 diopters [D]), corneal thickness > 480 μm, stable refraction for at least 1 year, no prior ocular surgery, and normal corneal topography. Exclusion criteria included keratoconus, systemic diseases affecting corneal biomechanics (e.g., diabetes), or incomplete follow-up data. SMILE$_{CXL}$ was recommended and performed for patients at higher risk of ectasia, including those with corneal thickness < 500 μm, spherical equivalent ≥ −8.00 D, or predicted residual stromal thickness < 300 μm.

## Surgical procedures

All surgeries were performed by a single experienced surgeon (Jiwon Jeong) under topical anesthesia (0.5% proparacaine hydrochloride) in a controlled environment (temperature 20–25°C, humidity 40–60%).

## SMILE procedure

The SMILE procedure utilized either the VisuMax 500 or VisuMax 800 femtosecond laser systems (Carl Zeiss Meditec, Jena, Germany) to create and extract a refractive lenticule within the corneal stroma, minimizing disruption to anterior layers. Patient docking used a curved contact glass interface to keep intraocular pressure (IOP) below 35 mmHg, avoiding discomfort or optic nerve strain. The laser ran at 500 kHz pulse frequency, with energy 100–125 nJ based on tissue response, spot and track spacing 3–4.0 μm for precise separation without excess heat. The VisuMax 800 offers a faster repetition rate (2 MHz compared to 500 kHz in VisuMax 500), reducing overall surgical time while maintaining similar precision and clinical outcomes. Lenticule parameters were customized by preoperative refraction: cap thickness 120 μm for residual stromal preservation, diameter 6.5 mm matching optical zone, and edge thickness ≥ 15 μm for handling. Scanning started with posterior lenticule surface in spiral-out pattern for depth uniformity, then anterior in spiral-in for correction, followed by side cuts. A 2 mm superior or temporal incision allowed access. Extraction used microforceps through the incision, separating lenticule carefully to avoid tears. Verification ensured complete removal without fragments affecting outcomes. This flapless method was selected for superior biomechanical retention over larger-incision techniques. Postoperative care included topical antibiotics (moxifloxacin 0.5%) for 3 days and steroids (fluorometholone 0.1%) tapered over 4 weeks, artificial tears, and follow-ups at 1 and 3 months.

## CXL procedure

In the SMILE$_{CXL}$ group, standard epi-on CXL was performed concurrently with SMILE following lenticule extraction to boost stability, using a modified Dresden protocol variant adapted for SMILE integration to reduce time while stiffening collagen effectively. This method used lenticule extraction for implicit epithelial removal, skipping separate debridement, and involved instilling Paracel solution (0.25% Riboflavin, HPMC, BAC, Tris) for 3 minutes on the corneal surface to break down epithelial tight junctions, followed by Vibe Xtra solution (0.22% Riboflavin, Saline, Isotonic) for 90 seconds to ensure saturation. UVA irradiation followed at 365 nm, 30 mW/cm$^2$ intensity for 90 seconds via calibrated Avedro system (Waltham, MA, USA), delivering 2.7 J/cm$^2$ fluence. Riboflavin was reapplied every 30 seconds during irradiation for hydration and dehydration protection.

## Biomechanical assessment

The BOSS (Intelon Optics, Woburn, MA, USA) scans were conducted in a controlled environment (temperature 22–24°C, humidity 40–60%) with patients fixating on a green LED target light to minimize eye movement artifacts. Each scan lasted approximately 5–7 minutes, acquiring Brillouin frequency shifts at 10 predefined points across the central 2.4-mm corneal zone: BM1 (nasal-central, x = 2.4 mm, y = 0 mm), BM2 (nasal-superior, x = 1.6 mm, y = 1.6 mm), BM3 (superior-central, x = 0 mm, y = 2.4 mm), BM4 (temporal-superior, x = −1.6 mm, y = 1.6 mm), BM5 (temporal-central, x = −2.4 mm, y = 0 mm), BM6 (temporal-inferior, x = −1.6 mm, y = −1.6 mm), BM7 (inferior-central, x = 0 mm, y = −2.4 mm), BM8 (nasal-inferior, x = 1.6 mm, y = −1.6 mm), BM9 (central-superior, x = 0 mm, y = 0.8 mm), and BM10 (central-inferior, x = 0 mm, y = −0.8 mm). The system employed a near-infrared laser (wavelength $\lambda = 780$ nm) coupled with a high-resolution spectrometer for non-contact measurements. BM, representing the longitudinal elastic modulus, was calculated as $M' = (\lambda^2 \Delta f)/(4n \rho)$, where $\Delta f$ is the Brillouin frequency shift (typically 7.5–8.5 GHz in healthy corneas), n is the refractive index (1.376), and $\rho$ is tissue density (1060 kg/m$^3$); this modulus correlates with viscoelastic properties, enabling quantification of stiffness independent of IOP. Intra-observer repeatability was high, with intraclass correlation coefficients exceeding 0.95 across repeated scans, as validated in prior evaluations to account for potential motion or blinking artifacts. Measurements were performed preoperatively and at 1 and 3 months postoperatively, with patients seated comfortably and each point scanned in triplicate; averages were computed for analysis, incorporating motion-tracking algorithms to enhance accuracy in dynamic in vivo conditions (Fig 1).

BM changes ($\Delta$BM) were calculated as postoperative BM minus preoperative BM for each point, with aggregate metrics (mean, minimum, maximum) derived to summarize overall trends (absolute BM values and $\Delta$BM). For exploratory regional insights, $\Delta$BM patterns were categorized descriptively based on sign ($\geq 0$ for stiffness preservation or enhancement, < 0 for loss) and inter-group statistical significance ($p < 0.05$ in at least one postoperative timepoint), yielding four categories with mean $\Delta$BM summaries to highlight clinical relevance, such as ectasia vulnerability. Spatial mapping visualized these patterns to identify corneal dynamic responses (e.g., nasal-temporal stabilization benefits from CXL).

## Statistical analysis

Continuous variables were expressed as mean ± standard deviation. Normality was assessed using the Shapiro-Wilk test. For between-group comparisons of BM values at 10 corneal points between the SMILE and SMILE$_{CXL}$ groups, an unpaired t-test was used for normality distributed data, while the Wilcoxon rank-sum test was applied for non-normal distributions. To address the violation of independence in the between-group comparisons of pre- and postoperative $\Delta$BM due to the inclusion of bilateral eyes from some participants, generalized estimating equations (GEE) with an exchangeable correction structure were used to account for within-individual correlations in the clustered data from both eyes. All statistical tests were two-sided, and a p value less than 0.05 was considered statistically significant. Data were analyzed using R software (version 4.4.0; R Foundation for Statistical Computing, Vienna, Austria).

## Results

### Demographics and refractive outcomes pre- and post-PSM

The pre-PSM SMILE$_{CXL}$ group included younger patients with greater myopia, thicker lenticules, and thinner residual stromal beds than the SMILE group, reflecting its use in higher-risk cases. Significant differences included age (26.9 ± 6.8 vs. 22.6 ± 6.0 years, p = 0.0003), LT (97.5 ± 27 vs. 125.1 ± 31 μm, p < 0.0001), RST (332.5 ± 33 vs. 295.8 ± 20 μm, p < 0.0001), and preoperative SE (−4.7 ± 2.0 vs. −6.9 ± 2.2 D, p < 0.0001) (Table 1).

Postoperative metrics also differed, with more residual myopia (SE at 3 months: −0.3 ± 0.5 vs. −0.7 ± 0.5 D, p = 0.0025) and flatter keratometry (Km at 3 months: 39.7 ± 2.0 vs. 38.0 ± 1.9 D, p = 0.0026) in SMILE$_{CXL}$, highlighting potential biases for biomechanical assessments.

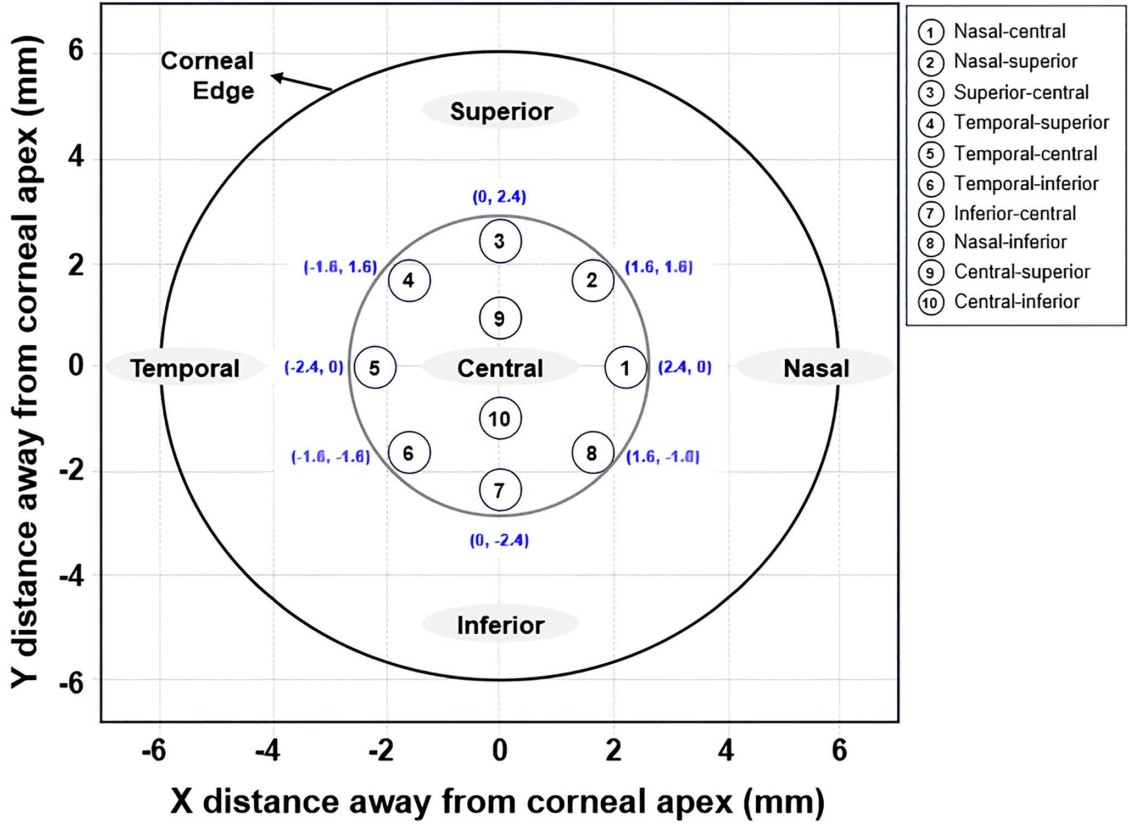

**Fig 1. Schematic representation of the 10 measurement points in the Brillouin Optical Scanning System (BOSS) with corneal limbus overlay.** BM1 (nasal-central: x = 2.4 mm, y = 0 mm); BM2 (nasal-superior: x = 1.6 mm, y = 1.6 mm); BM3 (superior-central: x = 0 mm, y = 2.4 mm); BM4 (temporal-superior: x = −1.6 mm, y = 1.6 mm); BM5 (temporal-central: x = −2.4 mm, y = 0 mm); BM6 (temporal-inferior: x = −1.6 mm, y = −1.6 mm); BM7 (inferior-central: x = 0 mm, y = −2.4 mm); BM8 (nasal-inferior: x = 1.6 mm, y = −1.6 mm); BM9 (central-superior: x = 0 mm, y = 0.8 mm); BM10 (central-inferior: x = 0 mm, y = −0.8 mm). Axes: X (nasal to temporal), Y (inferior to superior), −2.4 mm to +2.4 mm from apex (0,0); corneal edge (limbus) for anatomical reference.

PSM balanced these using 10 covariates (age, gender, LT, RST, SPH, CYL, SE, K1, K2, Km), yielding matched groups of 17 eyes each. Post-PSM, no significant differences remained (all p > 0.05), including age (23.2 ± 4.0 vs. 25.9 ± 6.9 years, p = 0.1743), LT (113.2 ± 24 vs. 106.9 ± 32 μm, p = 0.5166), RST (313.9 ± 28 vs. 306.6 ± 22 μm, p = 0.3981), preoperative SE (−5.8 ± 1.7 vs. −5.4 ± 2.1 D, p = 0.5034), and 3-month SE (−0.5 ± 0.7 vs. −0.7 ± 0.3 D, p = 0.5418) or Km (39.0 ± 2.0 vs. 39.3 ± 2.4 D, p = 0.8844), enabling focused evaluation of CXL's biomechanical effects (Table 1).

### Point-specific BM values

Preoperative BM values at the 10 corneal points showed no significant inter-group differences (all p > 0.05), with means of 2.867 ± 0.034 GPa (SMILE) and 2.858 ± 0.039 GPa (SMILE$_{CXL}$; p = 0.5082) (Table 2).

Postoperatively, the SMILE group showed BM reductions at most points at 1 month (e.g., BM1 from 2.859 ± 0.036 to 2.813 ± 0.057 GPa; BM5 from 2.860 ± 0.038 to 2.822 ± 0.062 GPa), indicating early stiffness compromise. By 3 months, values partially recovered but stayed below baseline in 8 points, with further reductions at BM7 (2.851 ± 0.064 to 2.840 ± 0.038 GPa) and BM10 (2.855 ± 0.059 to 2.848 ± 0.033 GPa). In contrast, the SMILE$_{CXL}$ group maintained or slightly increased values at all points. Significant inter-group differences occurred at 1 month for BM1 (p = 0.0159), BM5 (p = 0.0351), and minimum BM (p = 0.0318), and at 3 months for BM7 (p = 0.0492). Aggregate metrics confirmed SMILE's

**Table 1. Demographics and changes in refractive variables pre- and post-PSM.**

| Variables | | pre-PSM SMILE | SMILE$_{CXL}$ | p-value | post-PSM SMILE | SMILE$_{CXL}$ | p-value |
|---|---|---|---|---|---|---|---|
| Age | (year) | 26.9±6.8 | 22.6±6.0 | **0.0003** | 23.2±4.0 | 25.9±6.9 | 0.1743 |
| Patients | (number) | 161 | 19 | – | 16 | 10 | – |
| Gender ratio [male(%):female] | | 72(45%):89 | 7(37%):12 | 0.4360 | 6(38%):10 | 6(60%):4 | 0.1693 |
| Eye | (number) | 321 | 37 | – | 17 | 17 | – |
| LT | (µm) | 97.5±27 | 125.1±31 | **<.0001** | 113.2±24 | 106.9±32 | 0.5166 |
| RST | (µm) | 332.5±33 | 295.8±20 | **<.0001** | 313.9±28 | 306.6±22 | 0.3981 |
| SPH (D) | Pre | −4.1±1.9 | −6.0±2.0 | **<.0001** | −5.1±1.6 | −4.8±1.9 | 0.5687 |
| | Post 1M | −0.1±0.5 | −0.4±0.5 | **<.0001** | −0.1±0.7 | −0.2±0.4 | 0.5891 |
| | Post 3M | −0.1±0.5 | −0.6±0.6 | **<.0001** | −0.4±0.6 | −0.5±0.4 | 0.7539 |
| CYL (D) | Pre | −1.2±0.8 | −1.8±1.2 | **<.0001** | −1.4±0.7 | −1.2±0.9 | 0.5273 |
| | Post 1M | −0.4±0.3 | −0.4±0.3 | 0.9711 | −0.6±0.4 | −0.4±0.2 | 0.1052 |
| | Post 3M | −0.4±0.3 | −0.5±0.3 | 0.3875 | −0.6±0.4 | −0.5±0.3 | 0.2522 |
| SE (D) | Pre | −4.7±2.0 | −6.9±2.2 | **<.0001** | −5.8±1.7 | −5.4±2.1 | 0.5034 |
| | Post 1M | −0.2±0.5 | −0.6±0.5 | **0.0024** | −0.4±0.7 | −0.2±0.4 | 0.4637 |
| | Post 3M | −0.3±0.5 | −0.7±0.5 | **0.0025** | −0.5±0.7 | −0.7±0.3 | 0.5418 |
| K1 (D) | Pre | 42.4±2.7 | 42.4±1.2 | 0.8671 | 40.7±9.6 | 42.4±1.3 | 0.4887 |
| | Post 1M | 38.9±2.1 | 38.0±1.9 | **0.0115** | 38.8±1.8 | 38.6±1.9 | 0.7667 |
| | Post 3M | 39.0±2.0 | 37.6±1.6 | **0.0003** | 38.8±1.9 | 37.9±1.8 | 0.2004 |
| K2 (D) | Pre | 43.9±1.6 | 44.3±1.6 | 0.1621 | 44.9±1.3 | 44.0±1.7 | 0.0868 |
| | Post 1M | 39.8±2.2 | 38.8±1.9 | **0.0087** | 39.8±1.7 | 39.3±2.0 | 0.4146 |
| | Post 3M | 39.8±2.2 | 38.4±1.8 | **0.0004** | 39.9±1.8 | 38.8±2.0 | 0.1223 |
| Km (D) | Pre | 43.3±1.6 | 43.5±1.3 | 0.5091 | 44.0±1.5 | 43.3±1.5 | 0.1744 |
| | Post 1 M | 39.2±2.1 | 38.4±1.8 | 0.0566 | 39.0±1.3 | 39.3±1.9 | 0.6694 |
| | Post 3 M | 39.7±2.0 | 38.0±1.9 | **0.0026** | 39.0±2.0 | 39.3±2.4 | 0.8844 |

Note: PSM = propensity score matching; SMILE = small incision lenticule extraction; SMILE$_{CXL}$ = SMILE with corneal cross-linking; LT = lenticular thickness; RST = residual stromal thickness; SPH = spherical power; CYL = cylinder; SE = spherical equivalent; Km = mean keratometry; D = diopters. Categorical data (gender ratio) as frequencies/percentages; continuous data as mean±standard deviation. Analyses: independent t-tests (continuous variables). PSM used 10 factors (age, gender, LT, RST, SPH, CYL, SE, K1, K2, Km) based on significant inter-group differences across timepoints. Bolded p-values indicate significance (p<0.05). Dashes indicate no values in category.

decline (mean from 2.867±0.034 to 2.834±0.057 GPa at 1 month, recovering to 2.843±0.047 GPa at 3 months) vs. SMILE$_{CXL}$'s stability (2.858±0.039 to 2.860±0.028 GPa at 1 month, 2.862±0.028 GPa at 3 months), with min BM showing significant protection at 1 month. At 1 month, SMILE$_{CXL}$ yielded a mean stiffness increase of 0.52% (0.015 GPa), while SMILE alone showed a decrease of −0.50% (−0.014 GPa), indicating a 25–30% reduction in biomechanical weakening (Table 2). No adverse events were reported, though long-term follow-up is ongoing.

## Postoperative ΔBM changes

In the SMILE group, ΔBM values at 1 month postoperatively were negative in 9 of 10 corneal points (except BM10, which showed positive ΔBM), ranging from −0.044±0.070 GPa (BM1) to −0.009±0.061 GPa (BM7), with the greatest reductions at BM1 and BM5 (Table 3).

By 3 months, negative values persisted in 8 points (except BM4 and BM9, both ΔBM>0 GPa), spanning −0.028±0.065 GPa (BM1) to −0.002±0.035 GPa (BM3). In contrast, the SMILE$_{CXL}$ group showed ΔBM≥0 GPa across all points and timepoints, indicating complete stiffness maintenance. Inter-group analyses revealed significant differences at 1 month for

**Table 2. Comparison of BM values at 10 corneal points between SMILE and SMILE_CXL groups pre- and postoperatively.**

| | pre | | | post 1M | | | post 3M | | |
|---|---|---|---|---|---|---|---|---|---|
| **(GPa)** | **SMILE** | **SMILE_CXL** | **p-value** | **SMILE** | **SMILE_CXL** | **p-value** | **SMILE** | **SMILE_CXL** | **p-value** |
| **BM1** | 2.859 ± 0.036 | 2.850 ± 0.041 | 0.5180 | **2.813 ± 0.057** | 2.861 ± 0.024 | **0.0159** | 2.832 ± 0.053 | 2.861 ± 0.030 | 0.0709 |
| BM2 | 2.869 ± 0.036 | 2.863 ± 0.046 | 0.6911 | 2.839 ± 0.059 | 2.863 ± 0.035 | 0.1603 | 2.852 ± 0.039 | 2.851 ± 0.028 | 0.9116 |
| BM3 | 2.871 ± 0.046 | 2.863 ± 0.038 | 0.5879 | 2.833 ± 0.066 | 2.864 ± 0.029 | 0.1337 | 2.857 ± 0.032 | 2.870 ± 0.030 | 0.2543 |
| BM4 | 2.869 ± 0.025 | 2.856 ± 0.063 | 0.4591 | 2.845 ± 0.048 | 2.869 ± 0.035 | 0.1396 | 2.868 ± 0.034 | 2.864 ± 0.040 | 0.8182 |
| **BM5** | 2.860 ± 0.038 | 2.853 ± 0.039 | 0.5810 | **2.822 ± 0.062** | 2.863 ± 0.029 | **0.0351** | 2.834 ± 0.049 | 2.851 ± 0.034 | 0.2603 |
| BM6 | 2.868 ± 0.031 | 2.849 ± 0.035 | 0.1131 | 2.829 ± 0.061 | 2.852 ± 0.031 | 0.2087 | 2.845 ± 0.045 | 2.865 ± 0.032 | 0.1395 |
| **BM7** | 2.865 ± 0.039 | 2.868 ± 0.035 | 0.7896 | 2.851 ± 0.064 | 2.863 ± 0.027 | 0.5123 | **2.840 ± 0.038** | 2.864 ± 0.026 | **0.0492** |
| BM8 | 2.868 ± 0.047 | 2.859 ± 0.044 | 0.5583 | 2.842 ± 0.036 | 2.852 ± 0.030 | 0.4372 | 2.852 ± 0.042 | 2.868 ± 0.029 | 0.2001 |
| BM9 | 2.876 ± 0.044 | 2.867 ± 0.061 | 0.6187 | 2.844 ± 0.052 | 2.875 ± 0.038 | 0.0854 | 2.855 ± 0.035 | 2.863 ± 0.028 | 0.5052 |
| BM10 | 2.862 ± 0.051 | 2.857 ± 0.048 | 0.7576 | 2.855 ± 0.059 | 2.862 ± 0.030 | 0.6949 | 2.848 ± 0.033 | 2.864 ± 0.033 | 0.1760 |
| mean | 2.867 ± 0.034 | 2.858 ± 0.039 | 0.5082 | 2.834 ± 0.057 | 2.860 ± 0.028 | 0.1132 | 2.843 ± 0.047 | 2.862 ± 0.028 | 0.1606 |
| **min** | 2.841 ± 0.039 | 2.822 ± 0.047 | 0.2004 | **2.792 ± 0.060** | 2.830 ± 0.028 | **0.0318** | 2.818 ± 0.048 | 2.835 ± 0.028 | 0.2055 |
| max | 2.901 ± 0.038 | 2.893 ± 0.038 | 0.5211 | 2.876 ± 0.064 | 2.887 ± 0.034 | 0.5470 | 2.867 ± 0.043 | 2.886 ± 0.032 | 0.1490 |

Note: GPa = gigapascals; BM = Brillouin modulus; SMILE = small incision lenticule extraction; SMILE_CXL = SMILE with corneal cross-linking. Continuous data as mean ± standard deviation. Analyses: independent t-tests for inter-group comparisons. Bolded p-values indicate significance (p < 0.05), highlighting locations of BM reductions in SMILE and protection from CXL.

ΔBM1 (p = 0.0096), ΔBM4 (p = 0.0406), and ΔBM5 (p = 0.0144), and at 3 months for ΔBM1 (p = 0.0441), with trends toward protection elsewhere. Summary ranges across points: negative ΔBM in SMILE from −0.044 to −0.009 GPa (most to least negative) at 1 month and −0.028 to −0.002 GPa at 3 months; positive ΔBM in SMILE from 0.001 GPa (single value) at 1 month and 0.005 to 0.011 GPa at 3 months; in SMILE_CXL, positive ΔBM from 0.001 to 0.030 GPa at 1 month and 0.001 to 0.020 GPa at 3 months (no negatives). S1 Fig visualizes these via heat gradients (gray for SMILE losses, red for SMILE_CXL gains), with bolded p-values highlighting temporal group effects.

### Pattern-based categorization of ΔBM changes

To provide regional insights from the point-specific ΔBM values in Table 3, the 10 corneal points were categorized into four patterns based on temporal changes (1 and 3 months post-operation) and inter-group significance (p < 0.05 at least once), as summarized in Table 4.

In SMILE alone, most points showed early postoperative loss, persisting in 80% of points until 3 months, with some delayed worsening. In SMILE_CXL, ΔBM remained ≥0 across all points and timepoints, with significance highlighting zones of strong CXL efficacy (p < 0.05 for inter-group difference). The categories are: (1) Resistant but potentially Vulnerable (BM7 and BM10: initial near-zero loss post–SMILE but progressive worsening [greater BM loss at 3M than 1M], indicating ectasia

**Table 3. Comparison of BM changes (ΔBM) at 10 corneal points between SMILE and SMILE$_{CXL}$ groups pre- and postoperatively.**

| | ΔBM (GPa) | | | | | |
| --- | --- | --- | --- | --- | --- | --- |
| | Pre~post 1M | | | pre~post 3M | | |
| | SMILE | SMILE$_{CXL}$ | p-value | SMILE | SMILE$_{CXL}$ | p-value |
| **ΔBM1** | −0.044±0.070 | 0.016±0.042 | **0.0096** | −0.028±0.065 | 0.011±0.039 | **0.0441** |
| ΔBM2 | −0.020±0.061 | 0.013±0.065 | 0.1424 | −0.005±0.045 | 0.001±0.044 | 0.6997 |
| ΔBM3 | −0.018±0.076 | 0.020±0.050 | 0.1210 | −0.002±0.035 | 0.020±0.043 | 0.1351 |
| **ΔBM4** | −0.014±0.047 | 0.024±0.046 | **0.0406** | 0.011±0.043 | 0.014±0.044 | 0.8280 |
| **ΔBM5** | −0.040±0.076 | 0.018±0.046 | **0.0144** | −0.025±0.060 | 0.001±0.041 | 0.1725 |
| ΔBM6 | −0.034±0.065 | 0.002±0.056 | 0.1105 | −0.016±0.052 | 0.015±0.040 | 0.0640 |
| ΔBM7 | −0.009±0.061 | 0.018±0.046 | 0.1766 | −0.014±0.051 | 0.014±0.038 | 0.0986 |
| ΔBM8 | −0.016±0.035 | 0.001±0.057 | 0.3234 | −0.006±0.051 | 0.018±0.039 | 0.1383 |
| ΔBM9 | −0.012±0.056 | 0.030±0.053 | 0.0602 | 0.005±0.042 | 0.013±0.045 | 0.6388 |
| ΔBM10 | 0.001±0.038 | 0.005±0.057 | 0.8343 | −0.014±0.056 | 0.007±0.053 | 0.2730 |
| mean ΔBM (**min**, max) | −0.021±0.014 (−0.009, −0.044) | 0.015±0.009 (0.001, 0.030) | 0.0752 (**0.0111**, 0.3424) | −0.009±0.012 (−0.002, −0.028) | 0.011±0.006 (0.001, 0.020) | 0.1114 (0.0544, 0.1069) |

Note: GPa = gigapascals; BM = Brillouin modulus; SMILE = small incision lenticule extraction; SMILE$_{CXL}$ = SMILE with corneal cross-linking; ΔBM = pre- to post-operation change. Continuous data as mean ± standard deviation. Analyses: independent t-tests for inter-group comparisons. Bolded p-values indicate significance (p < 0.05), highlighting SMILE losses and SMILE$_{CXL}$ gains/preservation. Summary rows show ranges for negative ΔBM (decreases; most to least negative) and positive ΔBM (increases; min to max) across 10 points' means.

**Table 4. Pattern-based categorization of ΔBM changes over 3 months: Regional biomechanical responses in SMILE and SMILE$_{CXL}$.**

| No. | Pattern category | Points | Description | SMILE mean ΔBM (GPa) | | SMILE$_{CXL}$ mean ΔBM (GPa) | |
| --- | --- | --- | --- | --- | --- | --- | --- |
| | | | | (pre~1M) | (pre~3M) | (pre~1M) | (pre~3M) |
| 1 | Resistant but potentially vulnerable | BM7, BM10 | Initial resistance (near-zero/minimal loss) post-SMILE, but progressive worsening (greater BM loss at 3M than 1M), indicating ectasia risk. | −0.009(BM7), 0.001(BM10) | −0.014(BM7), −0.014(BM10) | 0.018(BM7), 0.005(BM10) | 0.014(BM7), 0.007(BM10) |
| 2 | Spontaneously and fully recovered | BM4, BM9 | Initial loss post-SMILE, but full spontaneous recovery (ΔBM ≥ 0 at 3M) without CXL, suggesting inherent resilience. | −0.014(BM4), −0.012(BM9) | 0.011(BM4), 0.005(BM9) | 0.024(BM4), 0.030(BM9) | 0.014(BM4), 0.013(BM9) |
| 3 | Moderately effective for CXL | BM2, BM3, BM6, BM8 | BM loss post-SMILE with partial spontaneous recovery (ΔBM < 0 at 3M), and moderate preservation via CXL (non-significant inter-group differences, p ≥ 0.05). | −0.020(BM2), −0.018(BM3), −0.034(BM6), −0.016(BM8) | −0.005(BM2), −0.002(BM3), −0.016(BM6), −0.006(BM8) | 0.013(BM2), 0.020(BM3), 0.002(BM6), 0.001(BM8) | 0.001(BM2), 0.020(BM3), 0.015(BM6), 0.018(BM8) |
| 4 | Highly effective for CXL | BM1, BM5 | Pronounced loss post-SMILE with partial spontaneous recovery (ΔBM < 0 at 3M), but strong preservation via CXL (significant inter-group differences, p < 0.05 at least once). | −0.044(BM1), −0.040(BM5) | −0.028(BM1), −0.025(BM5) | 0.016(BM1), 0.018(BM5) | 0.011(BM1), 0.001(BM5) |

Note: Patterns derived from spatiotemporal ΔBM trends at 1M and 3M post-operation (from Table 3), emphasizing regional vulnerabilities and CXL efficacy. Categories: (1) Resistant but potentially vulnerable (inferior-central zones); (2) Spontaneously and fully recovered (superior zones); (3) Moderately effective for CXL (peripheral zones); (4) Highly effective for CXL (nasal-temporal central zones). Data post-PSM; continuous values as mean ΔBM (GPa). GPa = gigapascals; BM = Brillouin modulus; SMILE = small incision lenticule extraction; CXL = corneal cross-linking; ΔBM = pre- to post-operation change; 1M/3M = 1/3 months.

risk); (2) Spontaneously and fully recovered (BM4 and BM9: initial loss post-SMILE but spontaneously full recovery without CXL [ΔBM ≥ 0 at 3M], suggesting inherent resilience); (3) Moderately effective for CXL (BM2, BM3, BM6, and BM8: BM loss post-SMILE with partial preservation via CXL, but non-significant differences [p ≥ 0.05]); (4) Highly effective for CXL (BM1 and BM5: pronounced loss post-SMILE but strong preservation with CXL, with significant differences [p < 0.05]).

## Spatial mapping of ΔBM patterns

Fig 2 illustrates the spatial distribution of the 10 BM points (matching Fig 1) overlaid with the four pattern categories from Table 4, within the central 2.4 mm × 2.4 mm corneal zone (−2.4 mm to +2.4 mm from apex). Categories are color-coded: resistant but potentially vulnerable (gray with diagonal hatching), spontaneously and fully recovered (white), moderately effective for CXL (yellow), and highly effective for CXL (orange). This mapping reveals zonal responses, such as robust CXL protection in nasal- and temporal-central regions (BM1/BM5), spontaneous recovery in superior areas (BM4/BM9), vulnerability in inferior zones (BM7/BM10), and moderate CXL effects in peripheral locations (BM2/BM3/BM6/BM8), showcasing BOSS's precision for personalized refractive strategies.

## Discussion

Our study demonstrates that CXL modestly enhances corneal biomechanical stability after SMILE, evidenced by preserved or slightly increased BM values and positive ΔBM trends over time. This protection is particularly evident in preventing early stiffness loss in high-risk patients with high myopia or thin residual stroma. By harnessing BOSS's high-resolution mapping, our findings highlight CXL's role in averting post-refractive ectasia while maintaining refractive outcomes. Through detailed point-specific analysis, temporal patterns, and spatial BM mapping, we uncover localized responses to SMILE with CXL, delivering fresh insights into regional stiffness preservation and its clinical impact.

Refractive surgeries, while effective for myopia correction, often weaken corneal biomechanics, risking ectasia (incidence 0.04–0.6%) from stromal disruption and shear stress in high-myopia or thin-cornea eyes [2,3]. LASIK induces notable stiffness losses (1.0–2.1%, or 0.03–0.06 GPa central reductions) as shown in multi-center studies using Brillouin

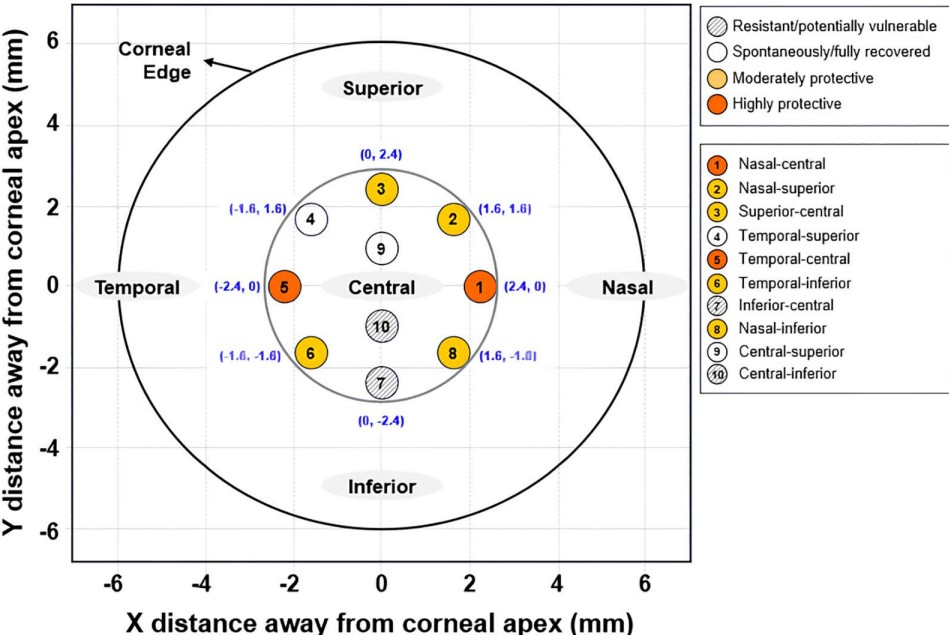

**Fig 2. Spatial mapping of biomechanical response categories at 10 corneal points over 3 months post-SMILE and SMILE$_{CXL}$.** Resistant but potentially vulnerable (gray: near-zero initial loss post-SMILE but progressive worsening [greater BM loss at 3M than 1M]; diagonal hatching for BM7/BM10); Spontaneously and fully recovered (white: initial loss post-SMILE with spontaneous recovery [ΔBM ≥ 0 at 3M] without CXL); Moderately effective for CXL (yellow: partial loss post-SMILE with non-significant CXL preservation [p ≥ 0.05]); Highly effective for CXL (orange: pronounced loss post-SMILE but significant preservation with CXL [p < 0.05]). Axes: X (nasal to temporal), Y (inferior to superior), −2.4 mm to +2.4 mm from apex (0,0). BM1–BM10 labels match Fig 1.

microscopy and other modalities like ocular response analyzers [14,15,17]. SMILE, being flapless, causes milder impacts (20–30% better stiffness retention than LASIK) due to preserved anterior stroma, with aggregate losses of ~0.7–1.2% confirmed in multi-cohort reviews and Brillouin studies [4,5,25,26]. Our PSM-matched results align with these generalizations, showing mean ΔBM of −0.021 GPa (−0.73%) at 1 month and −0.009 GPa (−0.31%) at 3 months, with incomplete recovery (80% points below baseline) and progressive inferior worsening (BM7/BM10), while enabling consistent comparisons through bias control [2,5,20]. Pronounced early losses at BM1/BM5 (−0.044 GPa [−1.54%], −0.040 GPa [−1.40%]; from Table 3) highlight focal vulnerabilities, amplifying ectasia susceptibility in weakest areas. These milder losses (~0.3–0.7% at 3 months) refine prior reports and position BOSS as a precise tool for detecting clinically relevant changes.

To mitigate refractive surgery-induced biomechanical weakening, CXL remains the primary strategy, enhancing collagen bonding via riboflavin-UVA to increase elastic modulus and degradation resistance [6,27]. Alternatives include accelerated CXL for efficiency and iontophoresis for non-invasive delivery, though the latter yields 10–20% less stiffening than standard epi-off CXL [28,29]. In refractive settings, CXL combined with LASIK/PRK achieves a stiffness increase of 0.5–1.0% (0.015–0.030 GPa), reducing biomechanical weakening by 15–25% and mitigating ectasia risk in high-myopia cases [7,14,15,28]. For SMILE, simulations indicate a 20–30% reduction in biomechanical weakening, equivalent to a stiffness increase of 0.3–0.5% (0.008–0.012 GPa), particularly benefiting thin corneas [4,5]. Our propensity score-matched SMILE$_{CXL}$ study demonstrates consistent biomechanical preservation across 10 measurement points, with a mean stiffness increase of +0.52% (0.015±0.009 GPa) at 1 month and +0.38% (0.011±0.006 GPa) at 3 months, contrasting with SMILE-alone declines of −0.73% (−0.021±0.014 GPa) at 1 month and −0.31% (−0.009±0.012 GPa) at 3 months (Tables 2 and 3). Notably, SMILE$_{CXL}$ maximally protected high-risk peripheral zones (BM1/BM5 at 1 month: +0.56%/+0.63%, 0.016/0.018 GPa, p<0.05) and progressive sites (BM7/BM10: maintaining ≥ 0%, despite SMILE-alone worsening), indicating CXL's independent collagen strengthening beyond natural recovery. This is evident in 8 points trending upward but remaining below baseline without CXL. These findings underscore CXL's standalone efficacy in SMILE, particularly for high-risk peripheral zones (BM1/BM5/BM7), advancing personalized SMILE$_{CXL}$ for patients with thin corneas or high myopia to prevent ectasia. Long-term tracking of recovery delays and durability remains essential to optimize outcomes.

Advances in corneal biomechanics rely on BOSS for high-resolution, non-invasive stiffness mapping via Brillouin shifts, as prior studies focused on aggregate metrics (e.g., central BM reductions of 0.03–0.06 GPa post-LASIK) in ectasia or post-refractive cases, revealing broad patterns but overlooking temporal dynamics and zonal heterogeneities [15,18,19,21]. This aligns with the recent review by Cao et al. [14], which positions Brillouin microscopy as an emerging tool for biomechanical analysis in ophthalmology, emphasizing its potential in detecting post-procedural changes such as those following corneal cross-linking; our study extends this framework by providing the first empirical, high-resolution spatiotemporal mapping specifically in SMILE and SMILE$_{CXL}$ contexts, addressing gaps in localized profiling for these procedures. For example, Brillouin spectroscopy detects asymmetry in early ectasia but lacks longitudinal integration to capture evolving risks like inferior weakening [16]. Comparative Brillouin analyses post-refractive surgery (PRK/LASIK/SMILE) note central reductions without detailed peripheral or time-resolved distinctions [23]. Our study overcomes these via a validated hierarchical spatiotemporal framework: point-specific ΔBM quantification (Table 3), temporal pattern categorization with inter-group significance (Table 4), and spatial mapping (Fig 2), ensuring robust, bias-minimized insights through PSM. This revealed four patterns: Resistant but potentially vulnerable (BM7/BM10: near-zero early change but worsening to −0.014 GPa at 3 months, indicating inferior ectasia risks [2,5,16]); Spontaneously and fully recovered (BM4/BM9: initial SMILE loss but recovery ≥ 0 GPa at 3 months without CXL, showing superior resilience; Moderately effective for CXL (BM2/BM3/BM6/BM8: SMILE reductions with non-significant CXL gains); Highly effective for CXL (BM1/BM5: pronounced SMILE losses countered by significant preservation, p<0.05 [14,28,29]). Deviating from uniform central dominance in prior reports, our PSM-balanced approach highlights SMILE's superior recovery and inferior vulnerabilities, positioning BOSS as essential for detecting clinically relevant changes [4,5]. Overall, this mapping uncovers heterogeneous BM changes post-SMILE (widespread initial weakening and inferior progression) vs. universal CXL preservation, refining

refractive biomechanics and supporting tailored strategies like targeted CXL for inferior risks or monitoring in high myopia to prevent ectasia and improve outcomes.

However, our study has several limitations. First, the retrospective, single-center design with one surgeon may introduce selection biases, though PSM on 10 covariates reduced confounders. Our results align with multi-center BOSS–LASIK studies (e.g., similar central gains post-CXL), suggesting broader applicability, but large-scale trials with diverse researchers are needed for generalizability. Second, the 3-month follow-up captures early stiffness changes but may miss long-term ectasia risks, given SMILE's incomplete recovery (80% points below baseline) and inferior worsening. Extending to 12–24 months, per ectasia protocols, would clarify stability and explore spontaneous recovery, CXL durability, and delays to inform healing variations and ectasia prediction in high-myopia cases. Third, lacking a CXL-only or ectasia-prone control limits isolating additive effects, though SMILE's limited recovery vs. CXL's broad boost suggests synergy. Randomized trials with risk-stratified groups, including a CXL-only arm in high-risk candidates, could confirm its impact and guide preventive use. Fourth, the modest BM shifts (0.01–0.04 GPa) raise questions on clinical relevance, addressable via correlations with outcomes like topography and visual acuity in larger cohorts. The observed small BM changes may translate to clinically meaningful outcomes, such as ectasia prevention or long-term refractive stability, by referencing studies showing that even modest stiffness increases correlate with reduced ectasia incidence in long-term follow-up [11,28]. Our observed shifts (0.01–0.04 GPa) align with similar scales in these studies (e.g., 0.017–0.035 GPa reductions post-LASIK correlating with biomechanical weakening), suggesting that such small BM variations may indicate early changes relevant for monitoring refractive outcomes. Fifth, the small post-PSM sample size (17 eyes/group) reduces statistical power and raises overfitting risk, though GEE addressed bilateral correlations. Larger cohorts would enhance robustness. Despite these limitations, they do not undermine the study's novelty as the first BOSS application to map localized stiffness in SMILE with CXL, offering insights into personalized refractive strategies. Furthermore, if future studies confirm CXL's safety and BM-preserving effects in normal corneas without irreversible adverse outcomes, it could extend applications beyond refractive surgery to populations with subclinical corneal weaknesses or early ectasia risks.

In conclusion, this BOSS-driven analysis reveals localized corneal stiffness dynamics post-SMILE and SMILE with CXL, highlighting CXL's key role in modestly countering biomechanical weakening–especially in vulnerable nasal-temporal and inferior zones–while supporting spontaneous superior recovery. By surpassing aggregate metrics with spatiotemporal patterning, our findings position BOSS as a tool for precision assessment of corneal biomechanics, contributing to evidence-based evaluations of refractive surgery outcomes.

## Supporting information

**S1 Fig. Heatmap of ΔBM changes between SMILE and SMILE$_{CXL}$ at 1 and 3 months post-operation.** Color gradients indicate: gray for losses (ΔBM < 0, mainly SMILE), red for preservation/gains (ΔBM ≥ 0, mainly SMILE$_{CXL}$). Rows: BM1–BM10 and aggregates (mean/min/max); bolded $p < 0.05$. Highlights CXL's protective effects (e.g., significant at BM1, BM5). Analyses: independent t-tests for inter-group comparisons ($p < 0.05$ exact). Data presented as mean (GPa). (TIF)

## Acknowledgments

We express gratitude to the Fatima Eye Clinic staff for their assistance in data collection and patient care.

## Author contributions

**Conceptualization:** Jiwon Jeong, Mincheol Bae.

**Data curation:** Hui June Kim.

**Formal analysis:** Dong Wook Kim, Younghee Kim.

**Methodology:** Jiwon Jeong, Mincheol Bae.

**Project administration:** Hui June Kim, Younghee Kim.

**Supervision:** Jiwon Jeong.

**Validation:** Dong Wook Kim.

**Visualization:** Dong Wook Kim, Younghee Kim.

**Writing – original draft:** Younghee Kim.

**Writing – review & editing:** Jiwon Jeong, Younghee Kim.

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
