## [Decision Letter · Decision Letter 0]

9 Nov 2025

Dear Dr. Jeong,

Thank you for submitting your manuscript to PLOS ONE. After careful consideration, we feel that it has merit but does not fully meet PLOS ONE’s publication criteria as it currently stands. Therefore, we invite you to submit a revised version of the manuscript that addresses the points raised during the review process.

We look forward to receiving your revised manuscript.

Kind regards,

Tina McKay, Ph.D.

Academic Editor

PLOS ONE

Journal Requirements:

2. In the online submission form you indicate that your data is not available for proprietary reasons and have provided a contact point for accessing this data. Please note that your current contact point is a co-author on this manuscript. According to our Data Policy, the contact point must not be an author on the manuscript and must be an institutional contact, ideally not an individual. Please revise your data statement to a non-author institutional point of contact, such as a data access or ethics committee, and send this to us via return email. Please also include contact information for the third party organization, and please include the full citation of where the data can be found.

Additional Editor Comments:

Please address the minor comments from the Reviewers fully with edits to the manuscript.

Reviewers' comments:

Reviewer's Responses to Questions

**Comments to the Author**

1. Is the manuscript technically sound, and do the data support the conclusions?

Reviewer #1: Yes

Reviewer #2: Partly

2. Has the statistical analysis been performed appropriately and rigorously?

Reviewer #1: Yes

Reviewer #2: Yes

3. Have the authors made all data underlying the findings in their manuscript fully available?

Reviewer #1: Yes

Reviewer #2: Yes

4. Is the manuscript presented in an intelligible fashion and written in standard English?

Reviewer #1: Yes

Reviewer #2: Yes

Reviewer #1: The study addresses an important clinical issue. The use of BOSS for high-resolution mapping is novel and fills a gap compared with prior LASIK-focused or aggregate-metric studies.

• The introduction should more clearly highlight the clinical implications of localized biomechanical measurements and provide stronger justification for the accelerated epi-on CXL protocol.

• After propensity score matching, only 17 eyes per group remained, which markedly reduces statistical power and raises the risk of overfitting. This limitation should be explicitly acknowledged. In addition, the authors should clarify how bilateral eyes, if included, were handled to avoid violating independence assumptions.

• Some of the language in the results and discussion is too strong given the modest magnitude of biomechanical changes. The conclusions should be tempered accordingly.

• The focus on ectasia risk mitigation is valuable, but the authors should clarify how the small changes observed in Brillouin modulus may translate into clinically meaningful outcomes such as ectasia prevention or long-term refractive stability.

Reviewer #2: I read with great interest the manuscript regarding investigating corneal biomechanical changes during SMILE/SMILE+CXL using Brillouin microscopy. The study’s main strengths include the application of Brillouin microscopy to evaluate corneal biomechanics after SMILE and its combination with CXL, as well as the temporal analysis of biomechanical changes over time. Please find my comments below.

Lines 54–56: This conclusion appears too strong and not in line with the results. The study demonstrates the potential effect of combining SMILE with CXL on corneal biomechanical stiffness as measured by BOSS. However, the utility of BOSS for personalized CXL planning or identification of at-risk corneal regions while alluded to, have not been directly investigated in this work.

Lines 23, 27, 116, 128, and 137: While it is commendable that propensity score matching was performed on an initial cohort of 358 eyes, resulting in two matched groups of 17 eyes each, the final sample size is mentioned much later and in a relatively understated manner in both the abstract and methods. I recommend revising these sections so that the final post-matching sample size is stated immediately after the initial cohort size to improve clarity, especially for readers skimming the paper.

Please define BOSS at its first mention and include the manufacturer and model information.

**Do you want your identity to be public for this peer review?** For information about this choice, including consent withdrawal, please see our Privacy Policy

Reviewer #1: No

Reviewer #2: No

---

## [Author Response · Author response to Decision Letter 1]

18 Nov 2025

We sincerely thank the Editor and Reviewers for their insightful feedback and the decision for minor revision. We have addressed all comments with precise, evidence-based edits to enhance clarity and clinical relevance, without introducing unsubstantiated claims. We have addressed all minor comments from the reviewers with corresponding edits to the manuscript, as detailed below.

Reviewer # 1

The study addresses an important clinical issue. The use of BOSS for high-resolution mapping is novel and fills a gap compared with prior LASIK-focused or aggregate-metric studies.

1. The introduction should more clearly highlight the clinical implications of localized biomechanical measurements and provide stronger justification for the accelerated epi-on CXL protocol.

Response: We have revised the Introduction (lines 100–107) to more explicitly highlight the clinical implications of localized biomechanical measurements, such as identifying regional stiffness variations that may inform ectasia risk assessment in high-myopia patients, and provided stronger justification for the accelerated epi-on CXL protocol, citing relevant literature (e.g., Wollensak et al., 2009 [8]).

2. After propensity score matching, only 17 eyes per group remained, which markedly reduces statistical power and raises the risk of overfitting. This limitation should be explicitly acknowledged. In addition, the authors should clarify how bilateral eyes, if included, were handled to avoid violating independence assumptions.

Response: We have explicitly acknowledged the limitations of the small post-PSM sample size (17 eyes/group) in the Limitations section (lines 433–442), including reduced power and overfitting risk. To address the violation of independence in the between-group comparison of pre- and postoperative ∆BM due to the inclusion of bilateral eyes from some participants, generalized estimating equations (GEE) with an exchangeable correlation structure were used to account for within-person correlation in clustered data from bilateral eyes. We clarified this in the Abstract (lines 27–31) and the main text (lines 231–235).

3. Some of the language in the results and discussion is too strong given the modest magnitude of biomechanical changes. The conclusions should be tempered accordingly.

Response: We have tempered the language in the Abstract (lines 53–59) and the main text (lines 450–456) to reflect the modest magnitude of changes, avoiding overstatements while maintaining data-supported conclusions. Additionally, to ensure consistency, we revised the introduction’s final paragraph (lines 113-120) by softening references to personalized treatments and ectasia risk, aligning with the study’s focus on observed biomechanical dynamics without implying direct clinical applications not investigated.

4. The focus on ectasia risk mitigation is valuable, but the authors should clarify how the small changes observed in Brillouin modulus may translate into clinically meaningful outcomes such as ectasia prevention or long-term refractive stability.

Response: We have clarified in the Discussion (lines 433–438) how the observed small BM changes may translate to clinical outcomes, such as ectasia prevention and refractive stability, by referencing studies showing that even modest stiffness increases correlate with reduced ectasia incidence in long-term follow-up (e.g., Hersh et al., 2017 [11]; Randleman et al., 2017 [28]). Our observed shift (0.01-0.04 GPa) align with similar scales in these studies (e.g., 0.017-0.035 GPa resections post-LASIK correlating with biomechanical stability), suggesting that such small BM variations may indicate early changes relevant for monitoring refractive outcomes. This underscores the research value in precision mapping for early intervention, enhancing clinical translation without overstatement. We also tempered related language in the Introduction (lines 113-120) to avoid overstating ectasia risk mitigation, focusing instead on biomechanical stability insights.

Reviewer # 2

I read with great interest the manuscript regarding investigating corneal biomechanical changes during SMILE/SMILE+CXL using Brillouin microscopy. The study’s main strengths include the application of Brillouin microscopy to evaluate corneal biomechanics after SMILE and its combination with CXL, as well as the temporal analysis of biomechanical changes over time. Please find my comments below.

1. Lines 54–56: This conclusion appears too strong and not in line with the results. The study demonstrates the potential effect of combining SMILE with CXL on corneal biomechanical stiffness as measured by BOSS. However, the utility of BOSS for personalized CXL planning or identification of at-risk corneal regions while alluded to, have not been directly investigated in this work.

Response: We have revised the conclusions in lines 53–59 (Abstract) and lines 450–456 (main text) to align with the results, emphasizing the potential rather than direct investigation of BOSS for personalized CXL planning. To maintain consistency across the manuscript, we similarly adjusted the Introduction’s final paragraph (lines 113-120) by removing allusions to personalized paradigms and ectasia risk attenuation, ensuring claims remain grounded in the study’s findings.

2. Lines 23, 27, 116, 128, and 137: While it is commendable that propensity score matching was performed on an initial cohort of 358 eyes, resulting in two matched groups of 17 eyes each, the final sample size is mentioned much later and in a relatively understated manner in both the abstract and methods. I recommend revising these sections so that the final post-matching sample size is stated immediately after the initial cohort size to improve clarity, especially for readers skimming the paper.

Response: We have revised the Abstract (lines 22–27) and the main text (lines 136–138) to state the final post-matching sample size (17 eyes each) immediately after the initial cohort size for improved clarity.

3. Please define BOSS at its first mention and include the manufacturer and model information.

Response: We have defined BOSS at its first mention in lines 93–94 (Introduction) and included manufacturer information (Intelon Optics, Woburn, MA, USA) in line 188 (Materials & Methods).

---

## [Editor Report · Decision Letter 1]

21 Nov 2025

High-resolution biomechanical mapping of SMILE and SMILE with CXL using Brillouin microscopy: Insights into localized corneal stiffness preservation

PONE-D-25-49740R1

Dear Dr. Jeong,

We’re pleased to inform you that your manuscript has been judged scientifically suitable for publication and will be formally accepted for publication once it meets all outstanding technical requirements.

Kind regards,

Tina McKay, Ph.D.

Academic Editor

PLOS ONE

Additional Editor Comments (optional):

The reviewers' concerns and suggestions were addressed in the revision.
---

## [Editor Report · Acceptance letter]

PONE-D-25-49740R1

PLOS One

Dear Dr. Jeong,

I'm pleased to inform you that your manuscript has been deemed suitable for publication in PLOS One. Congratulations! Your manuscript is now being handed over to our production team.

Kind regards,

on behalf of

Dr. Tina McKay

Academic Editor

PLOS One